green chemistry

N-doped, graphene, oxygen reduction reaction, mechanism, density functional theory

**Authors for correspondence:**
Yongjun Liu
e-mail: liuyj@scu.edu.cn
Jianjun Li
e-mail: jjli@scu.edu.cn

This article has been edited by the Royal Society of Chemistry, including the commissioning, peer review process and editorial aspects up to the point of acceptance.

# Density functional theory study of active sites on nitrogen-doped graphene for oxygen reduction reaction

Ping Yan[1], Song Shu[1], Longhua Zou[1], Yongjun Liu[1,2], Jianjun Li[1,2] and Fusheng Wei[1,2]

[1]College of Architecture and Environment, Institute of New Energy and Low Carbon Technology, and [2]National Engineering Research Center for Flue Gas Desulfurization, Sichuan University, Chengdu 610065, People's Republic of China

(iD) YL, 0000-0002-4906-561X

Oxygen reduction reaction (ORR) remains challenging due to its complexity and slow kinetics. In particular, Pt-based catalysts which possess outstanding ORR activity are limited in application with high cost and ease of poisoning. In recent years, nitrogen-doped graphene has been widely studied as a potential ORR catalyst for replacing Pt. However, the vague understanding of the reaction mechanism and active sites limits the potential ORR activity of nitrogen-doped graphene materials. Herein, density functional theory is used to study the reaction mechanism and active sites of nitrogen-doped graphene for ORR at the atomic level, focusing on explaining the important role of nitrogen species on ORR. The results reveal that graphitic N (GrN) doping is beneficial to improve the ORR performance of graphene, and dual-GrN-doped graphene can demonstrate the highest catalytic properties with the lowest barriers of ORR. These results provide a theoretical guide for designing catalysts with ideal ORR property, which puts forward a new approach to conceive brilliant catalysts related to energy conversion and environmental catalysis.

## 1. Introduction

Fuel cells, affording many attractive properties, like high energy conversion efficiency, low noise and wide reactant sources [1–3], are widely investigated as environmentally friendly and clean energy. Oxygen reduction reaction (ORR) is of importance for fuel cells but usually is slow under the nature condition due to slow kinetic properties [4–6]. It is urgent to seek ORR catalysts that can improve ORR performance. Pt-based materials are one

**Figure 1.** Charge density difference of characteristic N atoms on 1Gr (*a*), 3Py (*b*), 3Py_3Gr (*c,d*) and 2Gr (*e*). The yellow (blue) region denotes electron depletion (accumulation). All the isosurfaces are 0.10 e Å$^{-3}$. The grey and blue balls represent carbon and nitrogen atoms, respectively.

of the highest efficiency electrocatalysts for ORR [7–9] and are limited in their large-scale and commercial applications since they are rare, costly and poisoning [10–13]. Thus, searching for ORR electrocatalysts that supply properties such as low cost, high efficiency and stability is indispensable.

Graphene has good optical, electrical and mechanical characteristics, and is been proven to be the most promising non-metallic two-dimensional material in the twenty-first century [14–16], after it was first reported by Novoselov *et al.* [17] in 2004. In recent years, efforts have been made to use graphene to prepare ORR electrocatalysts via surface modification like heteroatom doping and other methods [18–20] to enhance activity and durability. Nitrogen-doped graphene, having good stability, advanced performance, easy preparation, high catalytic ORR ability and CO tolerance, is deemed to be a good feasible material that can replace noble metal catalysts [21–23].

However, what type of nitrogen species accelerates the ORR reaction is unclear and has many controversies. For instance, Wu *et al.* [24] proposed that the N-doped graphene they synthesized exhibited favourable electrochemical performance like long endurance and high activity for ORR, and their results suggested the pyridinic N (PyN) was inclined to be the most active N species to promote ORR proceeding. According to Lin *et al.* [25], the formation of graphitic N (GrN) on graphene was regarded as fatal for excellent ORR catalytic performance, demonstrating good properties like long-term stability and resistance to methanol crossover which are superior to those of the commercial Pt/C catalysts. The experiment of Thomas *et al.* [26] manifested the coexistence of PyN and GrN dedicated to the incremental ORR activities. Furthermore, Faisal *et al.* [27] reported rich PyN and GrN co-doped graphene could exhibit good characters for ORR.

In general, the ORR active mechanism on N-doped graphene remains indistinct, and there is a lack of appropriate means to investigate the relationship of the reaction process, rate-limiting step and active N atoms at the atomic level. Therefore, spin-polarized density functional theory (DFT) calculations were carried out to investigate the high ORR performance on N-doped graphene. The minimum energy pathways (MEPs) of comprehensive ORR catalytic processes on the five N-doped graphene materials were calculated to better understand the relationship between oxygen reduction activities and nitrogen doping types.

# 2. Results and discussion

## 2.1. Analysis of catalyst properties

Different N species may exhibit different electronic properties. The charge difference density (figure 1) of the characteristic N over one GrN-doped graphene (1Gr), three PyN-doped graphene (3Py), three PyN and three GrN co-doped graphene (3Py_3Gr), and double GrN-doped graphene (2Gr) were detected. The yellow (blue) region denotes electron depletion (accumulation). There were a large number of blue areas around the GrN of 1Gr indicating the GrN was a negative charge centre that could favourably attract electrophilic molecules, which matched with those of 3Py_3Gr and 2Gr. The

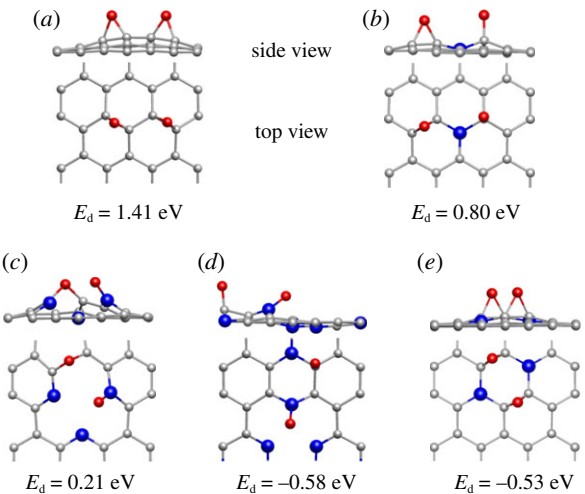

side view

top view

$E_d = 1.41$ eV    (a)    (b)    $E_d = 0.80$ eV

$E_d = 0.21$ eV    (c)    (d) $E_d = -0.58$ eV    (e) $E_d = -0.53$ eV

**Figure 2.** The optimal dissociative structures of $O_2$ on the five graphene materials. Where (a–e) stand for GP, 1Gr, 3Py, 3Py_3Gr and 2Gr, respectively. $E_d$ is the dissociative energy of $O_2$. Red balls represent oxygen atoms.

difference was that more yellow areas were rounded with the C atoms of 2Gr than those of 1Gr and 3Py_3Gr, illustrating there were more electrons lost for the C and stronger electronegativity for the GrN of 2Gr. Moreover, lots of blue areas surrounded the PyN of 3Py and 3Py_3Gr, and it presented a negative charge centre in the PyN similar to the GrN. Particularly, it was found that the charge density difference results of GrN and PyN on 3Py_3Gr were almost the same as those of the GrN on 1Gr and the PyN on 3Py, suggesting there might be no synergetic effects between the GrN and PyN which both existed on 3Py_3Gr.

Furthermore, Bader analysis (electronic supplementary material, table S1) was studied for the five graphene materials. As shown in electronic supplementary material, table S1, the charge values of the C atoms on pristine graphene (GP) were −0.05 and −0.08 e proven to be electrically neutral. The GrN and PyN on the other four N-doped graphene materials were given numerous electrons to be strongly electronegative accompanied by the amount of charge in the range of −1.10 e, which was consistent with the results of charge density difference. Those electronic analysis results denoted that introducing proper N atoms could redistribute the charge on the surface of graphene, facilitating charge transfer.

## 2.2. Identification of reaction pathways

### 2.2.1. $O_2$ adsorption on the catalysts

First, the optimal structures of $O_2$ adsorption on the five graphene materials were determined (electronic supplementary material, figure S1). It could be seen that the properties of $O_2$ adsorption on 1Gr, 3Py_3Gr and 2Gr were stronger than those on GP and 3Py, along with large adsorption energy. Besides, the $O_2$ adsorption energy of 2Gr was nearly equal to those of 1Gr and 3Py_3Gr which both involved the introduction of GrN. Additionally, the charge density difference and Bader analysis of adsorbed $O_2$ over the five graphene materials were further explored. The analyses of electron structure exhibited the charge transfer of the $O_2$ on 1Gr, 3Py_3Gr and 2Gr were more than those on GP and 3Py, and all of them were similar to each other. The result of electron structure was consistent with that of adsorption energy, which both figured that $O_2$ adsorption performances could be improved through the introduction of GrN while it could not be affected by the number of imports for GrN.

### 2.2.2. $O_2$ dissociation on the catalysts

Based on the optimal $O_2$ adsorption positions on the five graphene materials, $O_2$ dissociation was further carried out and dissociation positions were tested and the optimal constructions of dissociated $O_2$ on the five materials are displayed in figure 2. For GP material, the optimal site for $O_2$ dissociation was at the meta-position forming two epoxy groups, together with 1.41 eV dissociative energy. It required 0.80, 0.21 and −0.14 eV energy on 1Gr, 3Py and 2Gr to dissociate $O_2$, and severally formed an epoxy group and a C-O/PyN-O bond. Besides, C-O and PyN-O bonds were formed on 3Py_3Gr after $O_2$ dissociation, and the value of −0.58 eV energy was released. As a consequence, the order of $O_2$

dissociation energy on the five graphene materials was GP greater than 1Gr greater than 3Py greater than 2Gr greater than 3Py_3Gr.

What is more, the MEPs of $O_2$ dissociation on the five graphene materials were further figured out, manifested in detail in the electronic supplementary material, figures S2–S6. The energy barriers for $O_2$ dissociation on GP were more than 3.00 eV, that of 3Py was 2.00–3.00 eV and those of the three graphene materials including GrN-doped (1Gr, 3Py_3Gr and 2Gr) were in the range of 1.00–2.00 eV.

It is found that the lowest barrier of $O_2$ dissociation was required to be 1.08 eV which was too high to achieve, exhibiting that $O_2$ dissociating on the graphene materials are not feasible kinetics. Hence, $O_2$ tends to be in the molecular state participating in the ORR processes other than dissociated O atom. Hence, the MEPs of ORR processes were calculated in the following accompanied by $O_2$ in the molecular state.

## 2.3. Oxygen reduction reaction processes on the catalysts

According to the above determination of $O_2$ adsorption sites, the processes of $O_2$ reduction into $H_2O$ were examined by the minimum energy method. Relative potential energy and the side views of atomic structures for the initial state (IS), transition state (TS) and final state (FS) at each step for the five graphene materials are depicted in figure 3, and the detailed MEPs of $O_2$ reduction to $H_2O$ are shown by electronic supplementary material, figures S7–S11.

The initial hydrogenation of $O_2$ (denoted by 1H, and the following $n$ step is denoted as $n$H, $n = 1–4$) was the reaction rate-determining step for GP (figure 3a), and it needed a 0.35 eV energy barrier. The detailed analyses of 1H step on GP, 1Gr, 3Py and 3Py_3Gr were exhibited in our previous work [28], and they showed it was the most difficult for GP to initially hydrogenate $O_2$ and the easiest for 3Py_3Gr because of the sluggishness of the material and weak interaction between GP and $O_2$ and vice versa. As for 2Gr, the O-O bond activated into 1.26 Å, the changes of distance for H-O as well the O atom to 2Gr from IS to TS were little (electronic supplementary material, figure S11a). The favourable pre-activation of $O_2$ and the early arrival of TS for the 1H step on 2Gr were similar to those on 3Py_3Gr, attributed to very low activation energy (0.02 eV). The geometry results manifested that 2Gr had a good property for the 1H step of ORR processes like 3Py_3Gr.

The 2H step is forming O* and first $H_2O$ through hydrogenating OOH*. For 3Py, the 2H step was the rate-determining step for ORR processes, and a value of 0.27 eV was requested. Except for the high energy barrier on 3Py, the formation processes of 2H step on the other four materials were relatively easy, with the activation energy of 0.17 eV on GP (figure 3a), 0.02 eV on 1Gr (figure 3b), 0.04 eV on 3Py_3Gr (figure 3d) and 0.03 eV on 2Gr (figure 3e), respectively. The geometry analysis showed the distance of the H-O on 3Py was greatly reduced from IS to TS, with a larger change of 1.52 Å. But the activation of O-O was relatively weak with a slight change of 0.12 Å. There was a later advent of TS on 3Py compared with the other materials. Hence, a high activation barrier required on 3Py was used to overcome activating OOH* into $H_2O$.

It was found that the O* groups on the five graphene materials were all inclined to be hydrogenated into OH* groups, which adsorbed on the surface. In a word, the process of the 3H step occurred relatively easily for the five materials. An energy barrier of 0.11 eV was needed on GP and 0.01 eV on 3Py. Also, the 3H step was a spontaneous process on 1Gr, 3Py_3Gr and 2Gr.

As for 1Gr, 3Py_3Gr and 2Gr, hydrogenating OH* into $H_2O$ (4H step) was proven to be the reaction rate-limiting step. During the process of IS transferred to TS, the length of the H-O bond severally shortened by 1.37, 1.11, 1.39 and 1.37 Å for 1Gr (electronic supplementary material, figure S8d), 3Py (electronic supplementary material, figure S9d), 3Py_3Gr (electronic supplementary material, figure S10d) and 2Gr (electronic supplementary material, figure S11d), and the activation of O-O elongated into 0.17, 0.09, 0.16 and 0.15 Å. Although the O-O bonds on the four materials were activated, the emergence of a late TS led to relatively high energy should be conquered. However, the reaction went very well on GP (electronic supplementary material, figure S7d) due to an earlier arrival of TS and stronger activation of O-O (H-O from 2.92 to 2.00 Å, O-O from 1.51 to 1.58 Å), just requiring a small barrier.

For the five graphene materials, the catalytic activities of ORR were ranked as 2Gr > 1Gr > 3Py_3Gr > 3Py > GP. In contrast with 3Py_3Gr, 1Gr and 2Gr, a lower ORR property gains on 3Py despite incremental activity relative to that on GP. Besides, 3Py_3Gr denotes a lower ORR performance than 1Gr and 2Gr. It is considered that the introduction of GrN on 3Py_3Gr just changes the reaction rate-determining pathway and doesn't practically decrease the barrier while introducing PyN is the main reason for increasing ORR reactivity since its rate-limiting barrier is similar to that of 3Py. Moreover, 2Gr afforded the lowest energy barrier in the whole ORR processes compared with the other four

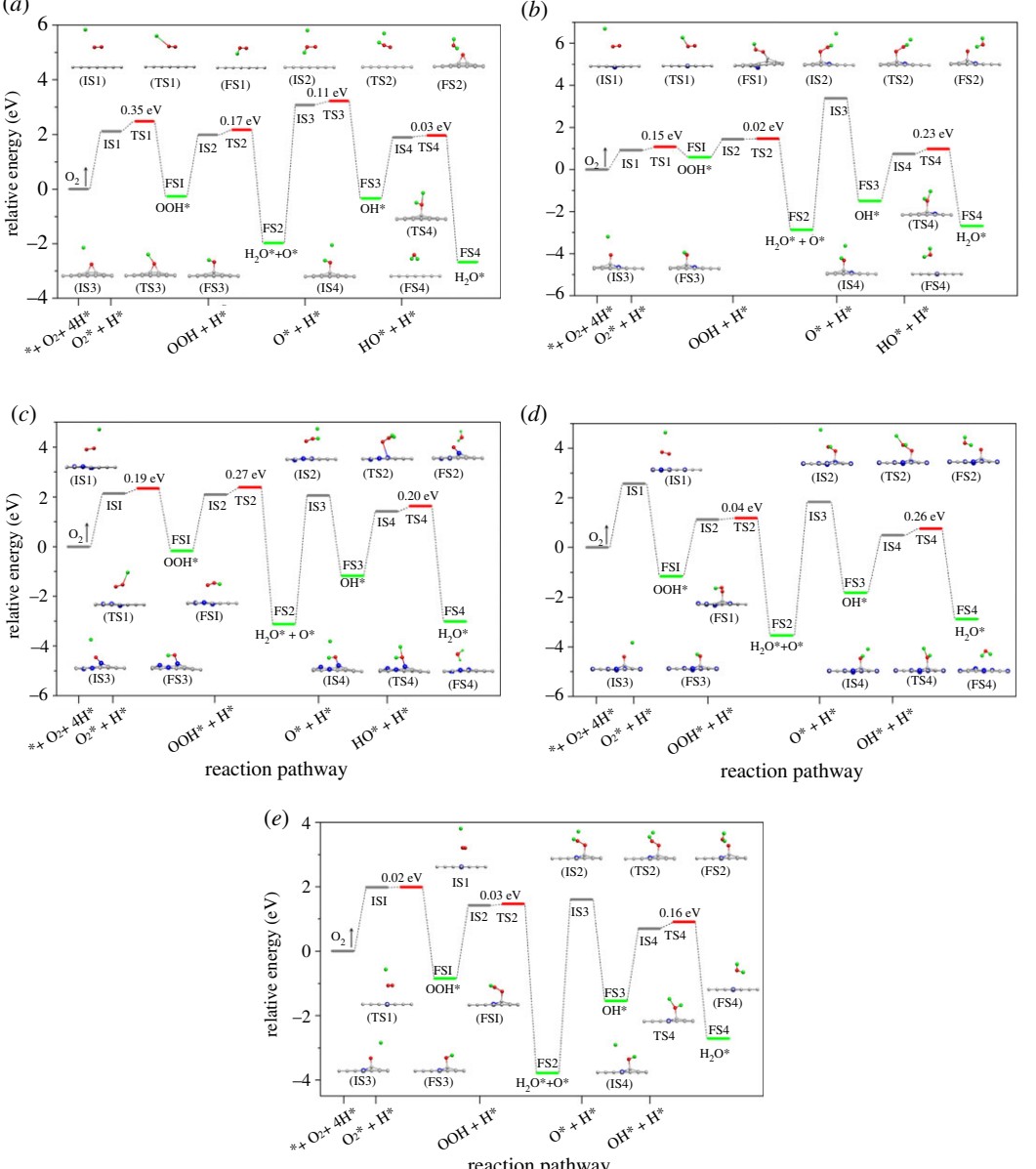

**Figure 3.** Reaction pathways for the ORR processes on GP (*a*), 1Gr (*b*), 3Py (*c*), 3Py_3Gr (*d*) and 2Gr (*e*). IS, TS and FS are initial state, transition state and final state, respectively. The * and $O_2$* express catalyst and the adsorbed $O_2$. The green balls stand for hydrogen atoms.

materials, indicating that N-doped graphene materials have good activities for ORR mainly ascribes the introduction of GrN, and the best doping type is dual-GrN.

## 2.4. The summary of the oxygen reduction reaction processes

Combined with the above results, the rate-limiting steps for the five graphene materials were summarized (figure 4). The rate-limiting step of ORR on GP was the 1H step initially protonating $O_2$, and the barrier was 0.35 eV. As for 3Py, an energy barrier of 0.27 eV on 3Py was required to be given to the rate-determining 2H step, the first $H_2O$ formation process. Besides, the rate-limiting step on 1Gr, 3Py_3Gr and 2Gr all were the 4H step that is the second $H_2O$ formation process, along with activation energy of 0.23, 0.26 and 0.16 eV, respectively.

Based on the rate-limiting step diagram of total ORR pathways on the five graphene materials, it could be gained that (i) compared with pristine graphene, the performances of ORR with $O_2$ in the molecular state are largely improved on the N-doped graphene materials, which is consistent with the previous results [29–31]; (ii) GrN denotes better $O_2$ adsorption and reduction performances compared

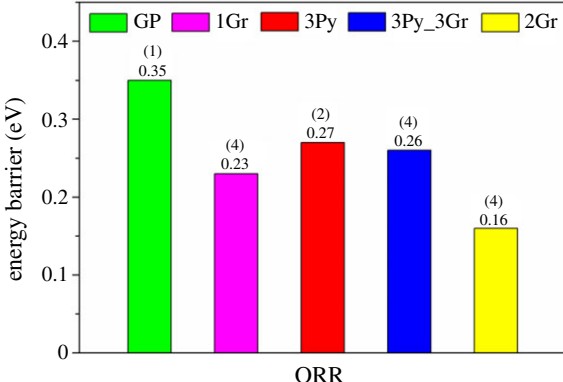

**Figure 4.** The rate-limiting step diagram for ORR on the five graphene materials. (1), (2) and (4) represent the reaction speed-determining steps are the first step, the second step and the fourth step, respectively. The energy barrier is in eV.

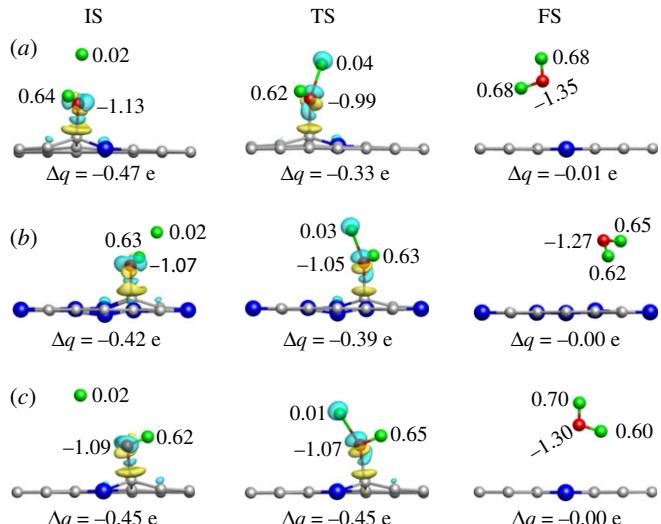

**Figure 5.** Charge density difference of IS, TS and FS at the 4H step for ORR on 1Gr (*a*), 3Py_3Gr (*b*) and 2Gr (*c*), respectively. The isosurfaces are 0.10 e Å$^{-3}$. $\Delta q$ represents the sum of the charge transfer numbers of the H* and OH*. The Bader value is in e.

with PyN; (iii) the co-introduction of PyN and GrN could not improve the ORR activity based on a single type of N dopant; and (iv) the double GrN para-doped graphene offers the best ORR performances among the five graphene materials. The results indicate that the main reason for high ORR activity on the N-doped graphene materials is the introduction of GrN.

# 3. Discussion

As mentioned above, 2Gr, which is equipped with the most outstanding properties for ORR among the graphene materials, is requested the maximum barrier at the 4H step. However, it referred that the other GrN-doped graphene materials (those are 1Gr and 3Py_3Gr), which exhibited good qualities at the first three steps of ORR, represent the worst performances at the 4H step. Thus, we focus the following discussion on 1Gr, 3Py_3Gr and 2Gr, revealing the reason why 2Gr can overcome the high barrier of the 4H step to express excellent characteristics for ORR processes but 1Gr and 3Py_3Gr do not.

First, the charge density difference and Bader analysis of IS, TS and FS for 1Gr, 3Py_3Gr and 2Gr at the 4H step were studied (figure 5). It was found that the FS that is reaction product H$_2$O, all were far away from 1Gr, 3Py_Gr and 2Gr, having scarcely any transferred electrons between the catalysts ($\Delta q \approx$ 0.00 e). The phenomena showed that the final resultant H$_2$O was easy to break away from the three GrN-doped graphene materials, so the catalytic material could continue to catalyse the reduction reaction, and it would not exist as accumulated product on the catalysts, which leads to poisoning on the catalyst.

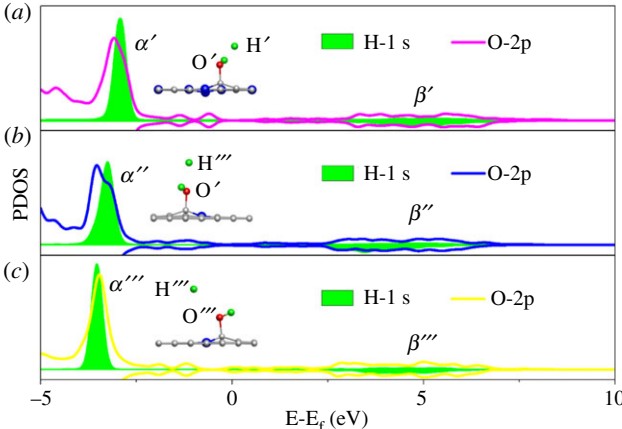

**Figure 6.** PDOS diagrams of the H and O atoms in IS of the 4H step on 1Gr (*a*), 3Py_3Gr (*b*) and 2Gr (*c*). The Fermi level is set to zero eV.

Furthermore, the charge distribution of the IS on 1Gr, 3Py_3Gr and 2Gr was similar, and the O and H atoms of the OH* group were strongly activated. Transferred charge of the introduced H atom in the TS slightly increased on 1Gr and 3Py_3Gr when that of the O and H atoms (belong to the OH*) slightly decreased compared with IS, which figured that the two materials both needed larger barriers to reach the TS due to the large changes and little activation. Besides, the 4H step on 1Gr was easier than that on 3Py_3Gr because the initial activation of the IS on 1Gr was stronger than that on 3Py_3Gr. Owing to greater activation in the IS and more transferred electrons number of the H* and OH* in the TS, 2Gr was found to show a higher activity than that of 1Gr and 3Py_3Gr, consistent with the former geometric analysis results. In a word, the charge density difference and Bader analysis express 2Gr have a better ORR activity than 1Gr and 3Py_3Gr.

Compared with 1Gr and 3Py_3Gr, the preferable activity of 2Gr at the 4H step can be further explained by the analysis of the partial density of states (PDOS). The PDOS of the O (for adsorbed OH* group) and introduced H atoms at the IS of the 4H step for 1Gr, 3Py_3Gr and 2Gr are shown in figure 6. According to the previous reports [28,32], the overlapping area of PDOS peaks between catalysts and the adsorbed molecule can be considered as an indicator of their interaction. It could be seen that the overlapping areas between the O-2p and H-1s orbitals for 2Gr were more than that for 1Gr and 3Py_3Gr, indicating there was a stronger effect between the O and introduced H atoms of 2Gr. Additionally, the H-1s and O-2p orbitals for 2Gr were coinciding at the position of $\alpha$ and $\beta$ peaks, and the peaks of the H-1s orbital were almost entirely wrapped in the O-2p orbital, those all resulted in there being intense interaction between the H and O atoms on 2Gr. As a result, the adsorbed OH* group on 2Gr could attract the introduced H atom more easily than that on 1Gr and 3Py_3Gr, manifesting 2Gr meets a higher performance for the 4H step of ORR and also for ORR.

## 4. Conclusion

In conclusion, by making use of spin-polarized DFT, comprehensive ORR processes on the five N-doped graphene materials were investigated. The results reveal that dissociating $O_2$ on the N-doped graphene materials is not favourable, and $O_2$ tends to participate in the ORR processes with the molecular state. Compared with pristine graphene, the ORR activities can be improved a lot by introducing the N atom. Generally, it is found that the introduction of GrN is beneficial for the adsorption and reduction of $O_2$. The co-introduction of GrN and PyN just overcomes the early high energy barrier step, but the reaction performances do not get substantially improved for the total ORR processes relative to a single introduction of GrN or PyN. It is nonexistent the synergistic effect between GrN and PyN to accelerate ORR proceeding. Furthermore, 2Gr manifests the more excellent ORR activities than the other four graphene materials, indicating that the high ORR electrocatalytic performances on the N-doped graphene materials attribute to the introduction of double GrN. Our works offer a novel mechanistic insight to understand the characteristics of N doping on the graphene materials for outstanding ORR performances and pave a new way for tailoring excellent catalysts in energy- and environmental-related catalytic reactions.

# 5. Models and methods

A periodical supercell of graphene including 72 C atoms was employed as a model catalyst with a rectangle boundary ($12.78 \times 14.76$ Å). A vacuum region of 20 Å was added perpendicular to the graphene plane to minimize the interaction between different layers [33]. Based on model graphene material that is pristine graphene (GP), four materials including one GrN-doped graphene (1Gr), three PyN-doped graphene (3Py), three PyN and three GrN co-doped graphene (3Py_3Gr) and double GrN-doped graphene (2Gr) were constructed (electronic supplementary material, figure S12). The detailed introductions about the first four catalysts are referred to as previous work [28,34,35]. Double GrN-doped graphene was added in this paper to systematically study the ORR activation mechanism on N-doped graphene. Six possible doping structures on double GrN-doped graphene materials were calculated (electronic supplementary material, figure S13), of which the most stable structure is the two GrN para-doped graphene (2Gr) exhibited in the electronic supplementary material, figure S13c.

The pathways of ORR are of complexity and multi-electron, multi-step reaction process, involving multiple intermediates with proton transfer. According to the different adsorption states of $O_2$, the ORR can be divided into two situations: $O_2$ in the molecular state and dissociative state take part in reactions. The total reaction can be denoted as follows and detailed formulae are presented in the electronic supplementary material.

$O_2$ in the molecular state

$$O_2 + 4H^+ + 4e^- \rightarrow 2H_2O, \tag{5.1}$$

$O_2$ in the dissociative state

$$O_2^* \rightarrow O^* + O^* \tag{5.2}$$

and

$$2O^* + 4H^+ + 4e^- \rightarrow 2H_2O. \tag{5.3}$$

All the first-principle spin-polarized DFT-D2 calculations were calculated by employing the VASP5.3 code [36,37], and the generalized gradient approximation with the Perdew–Burke–Ernzerhof (PBE) exchange and correlation functional [38]. A plane-wave basis set with cut-off energy of 400 eV within the framework of the projector-augmented wave (PAW) method [39,40] was used. The Gaussian smearing width was set to be 0.2 eV. The van der Waals (vdW) correction was described by the D2 method of Grimme with default parameters. The Brillouin zone was sampled with a $3 \times 3 \times 1$ Monkhorst Pack grid. All atoms, except those on the boundary, were allowed to relax and converge to $0.01\,\text{eV}\,\text{Å}^{-1}$ for all systems. The nudged elastic band (NEB) method [41,42] was used to search the MEP from an IS to its FS, and the TS was localized with the climbing image method and verified with a single imaginary frequency. Last but not least, the spin calculation and the solvent effect of water are considered in our paper.

The adsorption energy ($\Delta E_{\text{ads}}$) is defined as

$$\Delta E_{\text{ads}} = E_{\text{tot}} - (E_{\text{mole}} + E_{\text{sub}}), \tag{5.4}$$

where $E_{\text{tot}}$, $E_{\text{mole}}$ and $E_{\text{sub}}$ are the total energy of the adsorption complex, the isolated molecule and the GP substrate, respectively.

The dissociation energy ($\Delta E_{\text{d}}$) is defined as

$$\Delta E_{\text{d}} = E_{\text{d}} - (E_{\text{mole}} + E_{\text{sub}}), \tag{5.5}$$

where $E_{\text{d}}$ is the total energy of the dissociation complex.

Data accessibility. https://doi.org/10.5061/dryad.zgmsbcc9s.

Authors' contributions. Y.L. and J.L. designed the work and directed the experiments and supervised the project; P.Y. carried out the calculation study and wrote the paper; S.S. helped with sorting the paper; L.Z. assisted the simulation; F.W. co-supervised the project, and all authors discussed the results and commented on the manuscript.

Competing interest. There are no conflicts to declare.

Funding. This work was financially supported by Sichuan Provincial Science and Technology Agency Support Projects (nos. 2018KJT0027-2017SZ0170 and 2018GZ0414) and the Chengdu Science and Technology Bureau Huimin Projects (no. 2016-HM01-00075-SF). We thank the Institute of New Energy and Low Carbon Technology in SCU for computational service support.

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
