## [Peer Review File · Royal Society Open Science]

Review History

RSOS-210272.R0 (Original submission)

Review form: Reviewer 1

Is the manuscript scientifically sound in its present form?

No

Are the interpretations and conclusions justified by the results?

Yes

Is the language acceptable?

No

Do you have any ethical concerns with this paper?

No

Have you any concerns about statistical analyses in this paper?

No

Recommendation?

Major revision is needed (please make suggestions in comments)

Comments to the Author(s)

In this work the authors have methodically investigated ORR reaction for various N-doped graphene configurations. I have no concern regarding the calculations, results and the conclusions derived. However, I have a few queries and concerns

1. The manuscript needs serious grammatical and language editing. The language errors have made the manuscript incomprehensible and sometimes even misleading.
2. Notations such as PyN, GrN etc are not standard short-forms and hence they need to be explained prior to their use in manuscript.
3. There is no doubt that pyridinic N (PyN) and graphitic N (Gr-N) are important sites but I would like to know, why the oxidized N and pyrolic N sites are not included in the study?
4. The authors have studied five very specific N-doped graphene configurations but the reasons to specifically choose these configurations is not clear. Clarifying this would further emphasize the importance of the work done
5. The authors conclude that "...co-introduction of GrN and PyN overcomes the early high energy barrier step..." but they also state that synergistic effect between GrN and PyN are "non-existent". In which case, a reason for reduction of high energy barrier step due to co-existence of GrN and PyN needs to be provided.

Review form: Reviewer 2

Is the manuscript scientifically sound in its present form?

Yes

Are the interpretations and conclusions justified by the results?

Yes

Is the language acceptable?

Yes

Do you have any ethical concerns with this paper?

No

Have you any concerns about statistical analyses in this paper?

No

Recommendation?

Accept with minor revision (please list in comments)

Comments to the Author(s)

This theoretical calculation work on the research of ORR reaction efficiency is very good, and it provides thinking for the design of the catalyst in the experiment, but there are still several small issues that need to be explained.

1. The introduction part can continue to be optimized.
2. How does the authors confirm the rationality of the selected site?

Decision letter (RSOS-210272.R0)

Dear Dr Liu:

Title: DFT Study of Active Sites on Nitrogen-Doped Graphene for ORR
Manuscript ID: RSOS-210272

The editor assigned to your manuscript has now received comments from reviewers. We would like you to revise your paper in accordance with the referee and Subject Editor suggestions which can be found below (not including confidential reports to the Editor). Please note this decision does not guarantee eventual acceptance.

Please submit your revised paper before 13-May-2021. Please note that the revision deadline will expire at 00.00am on this date. If we do not hear from you within this time then it will be assumed that the paper has been withdrawn. In exceptional circumstances, extensions may be possible if agreed with the Editorial Office in advance. We do not allow multiple rounds of revision so we urge you to make every effort to fully address all of the comments at this stage. If deemed necessary by the Editors, your manuscript will be sent back to one or more of the original reviewers for assessment. If the original reviewers are not available we may invite new reviewers.

On behalf of the Subject Editor Professor Anthony Stace and the Associate Editor Dr Dattatray Late.

RSC Associate Editor:
Comments to the Author:
Major Revision needed

RSC Subject Editor:
Comments to the Author:
(There are no comments.)

Reviewers' Comments to Author:

Reviewer: 1

Comments to the Author(s)

In this work the authors have methodically investigated ORR reaction for various N-doped graphene configurations. I have no concern regarding the calculations, results and the conclusions derived. However, I have a few queries and concerns

1. The manuscript needs serious grammatical and language editing. The language errors have made the manuscript incomprehensible and sometimes even misleading.
2. Notations such as PyN, GrN etc are not standard short-forms and hence they need to be explained prior to their use in manuscript.
3. There is no doubt that pyridinic N (PyN) and graphitic N (Gr-N) are important sites but I would like to know, why the oxidized N and pyrolic N sites are not included in the study?
4. The authors have studied five very specific N-doped graphene configurations but the reasons to specifically choose these configurations is not clear. Clarifying this would further emphasize the importance of the work done
5. The authors conclude that "...co-introduction of GrN and PyN overcomes the early high energy barrier step..." but they also state that synergistic effect between GrN and PyN are "non-existent". In which case, a reason for reduction of high energy barrier step due to co-existence of GrN and PyN needs to be provided.

Reviewer: 2

Comments to the Author(s)

This theoretical calculation work on the research of ORR reaction efficiency is very good, and it provides thinking for the design of the catalyst in the experiment, but there are still several small issues that need to be explained.

1. The introduction part can continue to be optimized.
2. How does the authors confirm the rationality of the selected site?

Author's Response to Decision Letter for (RSOS-210272.R0)

See Appendix A.

RSOS-210272.R1 (Revision)

Review form: Reviewer 2

Is the manuscript scientifically sound in its present form?

Yes

Are the interpretations and conclusions justified by the results?

Yes

Is the language acceptable?

Yes

Do you have any ethical concerns with this paper?

No

Have you any concerns about statistical analyses in this paper?

No

Recommendation?

Accept as is

Comments to the Author(s)

Good

Decision letter (RSOS-210272.R1)

Dear Dr Liu:

Title: DFT Study of Active Sites on Nitrogen-Doped Graphene for ORR
Manuscript ID: RSOS-210272.R1

It is a pleasure to accept your manuscript in its current form for publication in Royal Society Open Science. The chemistry content of Royal Society Open Science is published in collaboration with the Royal Society of Chemistry.

On behalf of the Subject Editor Professor Anthony Stace and the Associate Editor Dr Dattatray Late.

RSC Associate Editor:
Comments to the Author:
Authors have revised the manuscript as per the reviewer's suggestion and now suitable for publication.

RSC Subject Editor:
Comments to the Author:
(There are no comments.)

Reviewer(s)' Comments to Author:
Reviewer: 2

Comments to the Author(s)
Good

Appendix A

Response Letter to the Editor of *Royal Society Open Science*

Dear Dr. Laura Smith,

We are very pleased to learn from your reports of revision for improvement of our manuscript entitled “*DFT Study of Active Sites on Nitrogen-Doped Graphene for ORR*” (Manuscript Number: RSOS-210272). Thanks for your patience and the helpful comments and suggestions of the referees.

To make the referee’s comments available, we have repeated the comments from the referee below (*in a different font*). We have responded to the comments point by point and have indicated the changes made in the revised manuscript (in blue color). Besides, we have checked through the whole manuscript carefully and some minor errors are corrected.

Response to Referee 1:

General Comment: In this work the authors have methodically investigated ORR reaction for various N-doped graphene configurations. I have no concern regarding the calculations, results and the conclusions derived. However, I have a few queries and concerns.

Response: Thanks a lot for the helpful suggestions.

Comment 1: The manuscript needs serious grammatical and language editing. The language errors have made the manuscript incomprehensible and sometimes even misleading.

Response: Thanks for this suggestion. We have carefully checked through the whole manuscript and Electronic Supplementary Information, and grammatical as well language problems have been revised.

Comment 2: Notations such as PyN, GrN etc are not standard short-forms and hence they need to be explained prior to their use in manuscript.

Response: Thanks for the critical comments and suggestions. This issue has been reviewed through the full manuscript and corrected.

Comment 3: There is no doubt that pyridinic N (PyN) and graphitic N (Gr-N) are important sites but I would like to know, why the oxidized N and pyrrolic N sites are not included in the study?

Response: Thanks for the carefulness and critical suggestion. Although oxidized N and pyrrolic N are important active sites, pyridinic N or graphitic N play an essential role in catalyzing ORR are the major controversy via researching numbers of ORR reported literature. To our knowledge, whether pyridinic N or graphitic N occupy the leading role for ORR is indistinct which is necessary to be investigated.

Comment 4: The authors have studied five very specific N-doped graphene configurations but the reasons to specifically choose these configurations is not clear. Clarifying this would further emphasis the importance of the work done.

Response: Thanks again. This study is based on our group's previous research seen in the "Models and methods" part, in other words, the five N-doped graphene configurations are based on the comprehensive consideration according to our previous literature research and studies.

Comment 5: The authors conclude that "...co-introduction of GrN and PyN overcomes the early high energy barrier step..." but they also state that synergistic effect between GrN and PyN are "non-existent". In which case, a reason for reduction of high energy barrier step due to co-existence of GrN and PyN needs to be provided.

Response: Thanks for this kindly and patient comment and suggestion. The reason why reduction of high energy barrier step due to co-existence of GrN and PyN has been explained in the first paragraph of "Analysis of catalyst properties" and the second paragraph of "ORR processes on the catalysts", which is to be the activity of the material and a strong interaction between 3Py_3Gr and O₂. Additionally, a detailed description of structural and electronic structures for the 1H step is seen in our previous work (it is referred to in the article), which is not our important part owing to 3Py_3Gr and the optimal ORR catalyst 2Gr whose rate-determined step is at the 4H step.

Response to Referee 2:

General Comment: This theoretical calculation work on the research of ORR reaction efficiency is very good, and it provides thinking for the design of the catalyst in the experiment, but there are still several small issues that need to be explained.

Response: Thanks a lot for the kind suggestions.

Comment 1: The introduction part can continue to be optimized.

Response: Thank you for the kind suggestion. The introduction part has been optimized. Besides, the abstract part also has been optimized.

Comment 2: How does the authors confirm the rationality of the selected site?

Response: Thanks for the comment. The selected site in this manuscript all has been tested and the minimum energy structure is confirmed to be the optimal structure.